# Bioengineered Cystinotic Kidney Tubules Recapitulate a Nephropathic Phenotype

**DOI:** 10.3390/cells11010177

**Published:** 2022-01-05

**Authors:** Elena Sendino Garví, Rosalinde Masereeuw, Manoe J. Janssen

**Affiliations:** Division Pharmacology, Utrecht Institute for Pharmaceutical Sciences, Utrecht University, Universiteitsweg 99, 3584 CG Utrecht, The Netherlands; e.sendinogarvi@uu.nl (E.S.G.); r.masereeuw@uu.nl (R.M.)

**Keywords:** nephropathic cystinosis, lysosomal storage disease, hollow fiber membrane, 3-dimensional models, autophagy

## Abstract

Nephropathic cystinosis is a rare and severe disease caused by disruptions in the *CTNS* gene. Cystinosis is characterized by lysosomal cystine accumulation, vesicle trafficking impairment, oxidative stress, and apoptosis. Additionally, cystinotic patients exhibit weakening and leakage of the proximal tubular segment of the nephrons, leading to renal Fanconi syndrome and kidney failure early in life. Current in vitro cystinotic models cannot recapitulate all clinical features of the disease which limits their translational value. Therefore, the development of novel, complex in vitro models that better mimic the disease and exhibit characteristics not compatible with 2-dimensional cell culture is of crucial importance for novel therapies development. In this study, we developed a 3-dimensional bioengineered model of nephropathic cystinosis by culturing conditionally immortalized proximal tubule epithelial cells (ciPTECs) on hollow fiber membranes (HFM). Cystinotic kidney tubules showed lysosomal cystine accumulation, increased autophagy and vesicle trafficking deterioration, the impairment of several metabolic pathways, and the disruption of the epithelial monolayer tightness as compared to control kidney tubules. In particular, the loss of monolayer organization and leakage could be mimicked with the use of the cystinotic kidney tubules, which has not been possible before, using the standard 2-dimensional cell culture. Overall, bioengineered cystinotic kidney tubules recapitulate better the nephropathic phenotype at a molecular, structural, and functional proximal tubule level compared to 2-dimensional cell cultures.

## 1. Introduction

Nephropathic cystinosis is an autosomal recessive chronic kidney disease condition caused by mutations in the *CTNS* gene [1,2]. Several mutations have been associated with this disease, but the most recurrent in Europe is a 57kb deletion including the first 10 exons of the *CTNS* gene [3]. This gene encodes for a cystine/proton symporter located in the lysosomal membrane. The impairment or loss of cystinosin leads to the accumulation of cystine inside the lysosomes in all the cells of the body [4,5,6], which causes severe and chronic damage to several organs, particularly the kidneys. One of the first manifestations of cystinosis is the clinical presentation of renal Fanconi syndrome, characterized by a severe proximal tubule cell dysfunction at early stages of the disease, which results in a total loss of integrity of the proximal tubule [7]. Great efforts have been made to elucidate further the underlying pathological mechanisms of nephropathic cystinosis that revealed several hallmarks beyond cystine accumulation, including impaired autophagy, mTOR activation, disrupted vesicle dynamics (lysosomes-autophagosomes interactions), mitochondrial impairment, reactive oxygen species (ROS), and increased cell stress [8,9,10,11,12,13]. Despite clinical improvements in prognosis, there is, as of yet, no curative therapy available for cystinosis. Therapy with cysteamine, the only treatment available, is symptomatic and its regime comprises of life-long drug intake with multiple reported side effects [14].

Most knowledge on the pathology of cystinosis results from in vitro models. Human fibroblast, obtained from cystinotic patients and widely available, are a common in vitro model which have been used for many years [15,16]. However, these cells lack the kidney phenotype and, therefore, their use for studying nephropathic cystinosis is limited. In vitro human proximal tubule models can be generated from fresh urine samples [17,18,19,20] and kidney tissue (biopsies) [21,22,23] from cystinotic patients. While valuable, their restricted availability and lack of capacity to stay in culture for more than a few passages urged researchers to immortalize primary cultures following different strategies. An example forms the ciPTEC (conditionally immortalized proximal tubule epithelial cells) [24], a model that has been shown to be particularly useful for investigating the molecular mechanisms affected in nephropathic cystinosis [9]. However, individual variability and the lack of a healthy control with the same genetic background as the cystinotic donor are major hurdles in pinpointing the molecular mechanism associated with the disease. To overcome this issue, recent studies have created isogenic cell lines using CRISPR-Cas by knocking out the *CTNS* gene [9,10]. In addition, advancements in the culture of primary cells obtained from patients’ urine now allow prolonged in vitro expansion in the form of adult organoids. This system also offers a 3-dimensional structure and a heterogeneous cell population that better recapitulate native kidney tissue and can be useful for disease modeling and personalized medicine [25]. Furthermore, the use of human induced pluripotent stem cells (hiPSC) has been presented as an unlimited source of patient specific material. Using this approach pluripotent stem cells can be obtained from any human tissue after de-differentiation and re-programming into kidney-like cells and organoids [26,27]. Coupling hiPSC and genome-editing tools such as CRISPR/Cas, can be used to create isogenic knock-out diseased cell lines from healthy hiPSC and, therefore, eliminate the variations that the differential genetic background between donors could carry [28].

Despite the advantages in human disease models for the proximal tubule, it remains challenging to replicate some of the features that the proximal tubule exhibits in nephropathic cystinosis, such as the total integrity loss of the nephron segment in renal Fanconi syndrome. We, therefore, aimed in this study to develop a cystinotic in vitro model that allows for perfusion and flow. The use of hollow fiber membranes (HFM) for the generation of microphysiological in vitro models has recently been described and has proven to be a promising in vitro platform for multiple organs that form tube-like structures, including kidney tubules and intestines [29,30,31]. The bioengineered kidney tubules consist of conditionally immortalized proximal tubule epithelial cells (ciPTEC) grown on stable and inexpensive biocompatible, poly-ether sulfone (PES) fibers [32]. The connection of the fibers to tubing and the application of flow allows for studying tubular epithelial integrity and transporter activity [32,33]. The ciPTEC line used for this work has been shown to stably express the proximal tubule phenotype over a very high number of passages, and when grown as kidney tubules on a fiber, these cells are capable of organic cation transporter (OCT)-mediated ASP+ transport, albumin reabsorption, organic anion secretion, and the secretion of immune modulators upon an inflammatory response [24,29,32,33].

In this study, we aimed to generate and characterize two cystinotic 3-dimensional models and to establish the added benefit of culturing cystinotic cells on HFM. We focused on the characterization of the cystinotic phenotype when cultured in these advanced in vitro platforms including epithelial monolayer organization and functional integrity. We hypothesize that the generation of advanced in vitro models that better reflect the pathophysiology of cystinosis in the proximal tubule is crucial to gain further knowledge about the molecular mechanisms underlying the tubular epithelium and functional disruption. This would also allow investigating novel treatment and potential curative options. 

## 2. Materials and Methods

### 2.1. Reagents and Antibodies

All reagents used were obtained from Sigma-Aldrich (Zwijndrecht, The Netherlands) unless specified otherwise. The primary antibodies and probes used for monolayer assessment were Phalloidin-AF488 (Invitrogen, Carlsbad, CA, USA) diluted 1:100, Mouse anti-α-tubulin (#EP1332Y; Invitrogen, Carlsbad, CA, USA) diluted 1:500 and rabbit anti-Na+/K+ATP-ase (a kind gift from Dr. Jan Koenderink, Radboudumc, The Netherlands) diluted 1:500. The primary antibodies used for autophagy and vesicle trafficking assessment were rabbit anti-LC3 (Novus Biologicals, Abingdon, UK #NB600-1384SS) diluted 1:1000, mouse anti-P62 (SQSTM1) (#610832; BD Biosciences, Mississauga, ON, Canada) diluted 1:1000, mouse anti-LAMP1 (#sc-18821; Santa Cruz Biotechnology, Dallas, TX, USA) diluted 1:200, and rabbit anti-mTOR (#2983; Cell Signaling Technology, Leiden, The Netherlands) diluted 1:400. The secondary antibodies used for detection were Polyclonal goat anti-rabbit (#P0448, Dako products, Carpinteria, CA, USA) diluted 1:5000, and polyclonal goat anti-mouse (#P0447, Dako products, Carpinteria, CA, USA) diluted 1:5000. Additionally, Alexa-488 goat anti- mouse (#ab150113; diluted 1:500), Alexa-647 goat anti-rabbit (#ab150083; diluted 1:200), donkey anti-rabbit (#AF647; diluted 1:300), and donkey anti-mouse (#AF568; diluted 1:200) secondary antibodies were all from Abcam (Amsterdam, The Netherlands). 

### 2.2. Cell Culture

Three ciPTECs lines were used for this study: the healthy control ciPTEC, *CTNS^WT^* (also referred to as ciPTEC14.4 in previous literature) [24], and two cystinotic models ciPTEC *CTNS^Patient^* (ciPTEC46.2) [34] and ciPTEC *CTNS*^−/−^. *CTNS*^−/−^ is an isogenic immortalized cell line derived from ciPTEC14.4 and generated in-house previously [9], which harbors a biallelic mutation in the exon 4 of the *CTNS* gene. The ciPTEC46.6 (referred to as “Patient”) is an immortalized cell line derived from a urine sample of a cystinotic patient harboring a 57 kb deletion which includes the first 10 exons of the *CTNS* gene. All ciPTEC were cultured as described previously [24]. Cells were seeded at a density of 48,400 cells/cm^2^ and allowed to grow at 33 °C for 24 h to enable proliferation and subsequently cultured at 37 °C for 7 days to mature them into fully differentiated PTECs. The culture medium used was Dulbecco’s modified Eagle medium DMEM/F-12 (GIBCO, Life Technologies, Paisley, UK) supplemented with fetal calf serum 10% (*v*/*v*), insulin 5 μg/mL, transferrin 5 μg/mL, selenium 5 μg/mL, hydrocortisone 35 ng/mL, epidermal growth factor 10 ng/mL, and triiodothyronine 40 pg/mL.

### 2.3. Hollow Fiber Membrane Culture

All cells were passaged and seeded on the HFM after reaching 80–90% confluency, as described previously [35,36]. MicroPES type TF10 hollow fiber capillary membranes (Membrana GmbH, Wuppertal, Germany) were cut into 175 cm pieces and sterilized in EtOH (70%, *v*/*v*) for 45 min. After sterilization, HFM were washed once with HBSS (Hanks’ Balanced Salt Solution, Gibco, Life Technologies, Paisley, UK) and put into sterile L-3,4-di-hydroxy-phenylalanine (L-Dopa, 2 mg/mL in 10 mM Tris buffer, pH 8.5) and incubated at 37 °C for 5 h. Next, HFM were washed with HBSS and incubated in a human collagen IV solution (25 μg/mL in HBBS) for 1 h at 37 °C. Unbound collagen IV was washed three times with HBSS. For cell seeding on the HFM, ciPTEC cells were washed and detached with accutase solution (Invitrogen, Carlsbad, CA, USA) and added to 1.5 mL Eppendorf tubes containing individual HFM at a density of 1 million cells/mL and incubated for 6 h at 37 °C, rotating the Eppendorf tubes 90° every 30 min. Lastly, seeded HFM were carefully placed into 6-well plates containing 3 mL of warm culture medium and allowed to grow for 3 days at 33 °C to facilitate the full coverage of the HFM by a cell monolayer, before transferring them to 37 °C for 7–10 days to obtained fully mature bioengineered kidney tubules. Medium was replaced every 3 days.

### 2.4. Immunostainings

Cells and kidney tubules were fixed using 4% PFA solution (Pierce™ 16% formaldehyde (*w*/*v*), methanol-free, ThermoFisher, Waltham, MA, USA) and subsequently permeabilized in 0.3% (*v*/*v*) Triton X-100 in HBSS. Kidney tubules were incubated with a blocking solution (2% (*w*/*v*) bovine serum albumin (BSA) fraction V and 0.1% (*v*/*v*) Tween-20 in HBSS) to prevent the non-specific binding of antibodies. Next, kidney tubules were incubated with the corresponding primary antibodies diluted in blocking solution overnight at 4 °C on a rocking platform. After washing off the primary antibodies, incubation with the secondary antibodies at RT for 2h was followed. Lastly, cells were incubated with Hoechst 33342 (dilution 1:10000) and kidney tubules were mounted using Prolong gold containing DAPI (Cell Signaling Technology, Leiden, The Netherlands) for nuclei staining. Images were acquired using the confocal microscope Leica TCS SP8 X (Leica Biosystems, Amsterdam, The Netherlands).

### 2.5. FITC-Inulin Leakage Assay

To quantify the tightness of the kidney tubules cell monolayer, a FITC-inulin solution (0.1 mg/mL in HBSS) was prepared. The kidney tubules were connected to a custom-made 3D-printer chamber [31] made of cytocompatible polyester after washing once with HBSS and sterilizing the chambers with EtOH 70%. The HFM were subsequently connected to a cannula (inner diameter 120–150 μm) DMT Trading, Aarhus, Denmark) using microsuture silk (Pearsalls Limited, Taunton, UK). The apical compartment of the chamber was filled with 1 mL of HBSS, and the kidney tubules were perfused with HBSS. Once leakage of the chamber was resolved, the kidney tubules were perfused with FITC-inulin solution at a rate speed of 0.1 mL/min for 10 min using a Terumo Syringe Pump TE-311 (Terumo, Leuven, Belgium). After completing the perfusion, three technical replicate samples of 100 uL were taken from the apical chamber and transferred to a 96 well-plate. Fluorescence was measured using a Tecan infinite M200PRO plate reader (Tecan Austria GmbH, Grödig, Austria) at an excitation and emission wavelength of 492 nm and 518 nm, resp.

### 2.6. Isolation of mRNA and Quantification by Real-Time PCR

To be able to obtain enough mRNA for a reliable gene expression measurement, three kidney tubules were pooled together in a 1.5 mL Eppendorf tube. Cell pellets were collected by detaching the cells from the HFM using accutase solution (Invitrogen, Carlsbad, CA, USA) neutralized with culture medium and centrifuged for 10 min at 300× *g*. The mRNAs were extracted using the RNeasy mini kit (Qiagen, Venlo, The Netherlands) following the manufacturer’s instructions. A total of 600 ng of mRNA was reverse transcribed using iScript Reverse Transcriptase Supermix (Bio-Rad Laboratories, Hercules, CA, USA). Lastly, quantitative real-time PCR was performed using iQ Universal SYBR Green Supermix (Bio-Rad Laboratories, Hercules, CA, USA) using RPS-13 (ribosomal protein subunit 13) as a reference gene for normalization. Relative gene expression levels were calculated as fold changes using the 2^−ΔΔCt^ method. Primers were designed using the free access online tool: https://ncbi.nlm.nih.gov/ (accessed on 1 February 2021) and ordered from ThermoFisher (Waltham, MA, USA). The primers used are shown in Table 1.

### 2.7. Lysosomal Cystine Measurement

Lysosomal cystine was quantified using HPLC-MS following an optimized assay previously developed in-house [37]. In short, six kidney tubules were pooled together in 1.5 mL. Eppendorf tubes and washed with ice-cold HBSS. Cells were then detached with accutase solution (Invitrogen, Carlsbad, CA, USA) and resuspended in HBSS before centrifugation at 300× *g* for 10 min. Cell pellets were neutralized with N-Ethylmaleimide (NEM) solution (5 mM NEM in 0.1 mM sodium phosphate buffer pH 7.4) to avoid the unwanted measurement of cytosolic cystine. Cell suspension was precipitated, and protein was extracted by adding sulfosalicylic acid 15% (*w*/*v*) and centrifuging at 20,000× *g* for 10 min at 4 °C. In parallel, the total protein quantification was assessed by the PierceTM BCA protein assay kit (Thermo Fischer, Waltham, MA, USA) following the manufacturer’s instructions. Lysosomal cystine measurement was performed using HPLC-MS/MS. Data is expressed as the lysosomal cystine normalized by total protein content.

### 2.8. Metabolomics Profiling

To be able to obtain enough metabolites for a reliable measurement, six kidney tubules were pooled in a 1.5 mL Eppendorf tube and washed with ice-cold HBSS. Next, cells pellets were collected by detaching the cells from the HFM using accutase solution (Invitrogen, Carlsbad, CA, USA), neutralized with culture medium, and centrifuged for 10 min at 300× *g*. Cell pellets were then incubated for 1 min with 1 mL of lysis buffer (methanol/acetonitrile/dH2O at a 2:2:1 ratio), vortexed for 30 s with intervals of 30 s in ice for 5 min, and put in a shaker platform at 4 °C for 20 min. The suspension was centrifuged at 16,000× *g* for 20 min at 4 °C and, lastly, supernatants containing the metabolite suspension were collected and stored at −80 °C until LC-MS measurement was performed. In parallel, six coated but unseeded HFM were pooled and identically processed as the seeded HFM, as the negative control. Additionally, medium samples of each well with unseeded HFM were cultured and identically processed as the other samples, as additional controls. All samples were sent to the Metabolism Expertise Center (Utrecht University, Utrecht, The Netherlands) and analyzed as described before [9]. In short, LC-MS analysis was performed on an Exactive mass spectrometer (ThermoFisher Scientific, Waltham, MA, USA) coupled to a Dionex Ultimate 3000 autosampler and pump (ThermoFisher Scientific, Waltham, MA, USA). Metabolites were separated using a Sequant ZIC-pHILIC column (2.1 cm × 150 mm, 5 μm, guard column 2.1 cm × 20 mm, 5 μm; Merck) with elution buffers acetonitrile (A) and eluent B (20 mM (NH_4_)_2_CO_3_, 0.1% NH_4_OH in ULC/MS-grade water (Biosolve, Valkenswaard, The Netherlands)). Gradient ran from 20% eluent B to 60% eluent B in 20 min, followed by a wash step at 80% and equilibration at 20%. Flow rate was set at 150 μL/min. Analysis was performed using the TraceFinder software (ThermoFisher Scientific, Waltham, MA, USA). Metabolites were identified and quantified based on exact mass within 5 ppm and further validated by concordance with retention times of standards. Data was further analyzed with R studio using the publicly available code from MetaboAnalyst. 

### 2.9. Statistical Analysis

Every experiment was performed in at least three biological replicates, including at least 3 technical replicates each, unless specified otherwise. Results are shown as the mean ± standard error of the mean (SEM). All statistical analyses, except for metabolomics and image analysis, were performed in GraphPad version 8 (GraphPad software, La Jolla, CA, USA), using one-way ANOVA followed by a Tukey’s post-hoc test, and two-way ANOVA for multiple comparison analyses. Metabolomics analysis was performed using R-studio (version 1.4.1103-3), using the publicly available code from MetaboAnalyst online tool. Image analysis was performed using Fiji (ImageJ software version 1.49, National Institutes of Health, Bethesda, MD, USA). For the actin filament orientation analysis, the plug-in “directionality” was used. For the puncta analysis, single channel images were first converted to binary and corrected for background. The puncta were counted by setting the same the threshold for all samples (between 30–255 and 3–255 for LC3/p62 and LAMP1/mTOR, respectively). Lastly, the plug-in “analyze particles” was used to count the particles. A *p*-value of <0.05 was considered as statistically significant.

## 3. Results

### 3.1. Healthy and Cystinotic ciPTECs Form Mature Kidney Tubules When Cultured on HFM

To develop a 3-dimensional bioengineered model of nephropathic cystinosis, we cultured different ciPTEC lines on HFM. We used the ciPTEC *CTNS^WT^* cells as our healthy control line; these cells have been well characterized in the past and are known to maintain many proximal tubules cell functions in culture [24]. The ciPTEC *CTNS*^−/−^ cell line has been obtained by knocking out *CTNS* from the *CTNS^WT^* parent cell line [9]. These cell lines are isogenic and therefore any differences seen between these lines then can be directly attributed to *CTNS* loss. In addition, we include the *CTNS^Patient^* ciPTEC line, which was derived from cystinotic patient harboring a 57kb deletion and presents a strong cystinotic phenotype. 

To determine whether all three ciPTEC lines can grow into a functional monolayer on the HFM, the polarization of the barrier was evaluated by immunofluorescent staining of the apically expressed cilia and the basolaterally located Na^+^/K^+^-ATPase. Imaging results confirmed that all three cell lines were able to attach and grow for at least 15 days on the double-coated HFM and exhibit one cilium at the apical side, and express Na^+^/K^+^-ATPase at the basolateral side of each cell (Figure 1A–C)

### 3.2. Bioengineered Cystinotic Kidney Tubules Present Disrupted Epithelial Monolayer

To assess the organization of the cell monolayer, kidney tubules were evaluated by immunofluorescent stainings and quantification of the intracellular actin filaments using the fluorescently labeled phalloidin probe. Imaging results revealed a loss of organization of the actin filaments in both cystinotic models when compared to the healthy proximal tubule model (Figure 2A–C). Moreover, directionality image analysis revealed that actin filaments were mostly oriented in a 90° angle (relative to the length of the tubule) in the healthy kidney tubules, an organization that was lost in the cystinotic models (Figure 2D–F). Besides, both cystinotic models exhibited holes along the monolayer where cells detached from the fiber membrane, consistent with a weaker and leakier monolayer. 

Next, we compared the phenotype of the 3D bioengineered kidney tubules to a 2D environment. The same cell lines used for the kidney tubules were seeded under the same conditions as in the HFM, on top of a layer of L-dopa and a layer of collagen IV to ensure that the results are not dependent on the extracellular matrix provided (Figure 3). Quantification of the phalloidin staining showed no organization of the actin filaments in either of the models (Figure 3), including the healthy control, indicating that the directional cell monolayer organization is due to the 3-dimensional architecture of the HFM.

To further assess the integrity and the tightness of the kidney tubules monolayer, a FITC-inulin leakage assay was performed (Figure 4A). The results confirmed that the monolayer of both cystinotic kidney tubule models is less tight in comparison to the healthy control, with the cystinotic models 2.5- (*CTNS*^−/−^) and 3.7-fold (*CTNS^Patient^*) leakier than the healthy control (Figure 4B).

### 3.3. Cystinotic Kidney Tubules Accumulate Cystine Due to Cystinosin Absence 

To evaluate the key phenotypical features of renal cystinosis, the expression of the *CTNS* gene and the accumulation of cystine in the lysosomes of the kidney tubules were measured. Real-time PCR results show that the *CTNS* gene expression was reduced by 60% in the *CTNS*^−/−^ kidney tubules and a total loss of *CTNS* expression in the *CTNS**^Patient^* tubules compared to the healthy kidney tubules (Figure 5A). The loss of *CTNS* expression led to a 100- and 360-fold increase in cystine accumulation in the *CTNS*^−/−^ and *CTNS**^Patient^* kidney tubules, respectively (Figure 5B).

### 3.4. Intracellular Vesicle Trafficking Is Impaired in Cystinotic Kidney Tubules

We evaluated further the downstream effects of lysosomal cystine accumulation by assessing autophagy using LC3 and p62 protein levels, which are part of the autophagosomal membrane and autophagosome cargo, respectively. Image analyses revealed a significant accumulation of both LC3 (5-fold in *CTNS*^−/−^ and 8-fold in *CTNS**^Patient^* kidney tubules) and p62 (2-fold in *CTNS*^−/−^ and 2.6-fold in *CTNS**^Patient^* kidney tubules) in the cytosol of the cystinotic models (Figure 6A–D), indicating increased levels of autophagy in the cystinotic kidney tubules. Real-time PCR quantification show the dysregulation of *TFEB* and *SQSTM1* (the gene coding for p62) in both cystinotic kidney tubules when compared to baseline healthy kidney tubules expression levels. *TFEB* is a transcriptional regulator of autophagy which is known to downregulate its own expression after autophagy activation. Indeed, its expression appeared significantly reduced by 40% to 80% in *CTNS*^−/−^ and *CTNS**^Patient^* kidney tubules, respectively (Figure 6E). Interestingly, mRNA expression of p62 was 2-fold higher in *CTNS*^−/−^ kidney tubules than the healthy control, while in the *CTNS**^Patient^* kidney tubules, a 44% reduction in p62 gene expression was found (Figure 6E).

To confirm the activation of autophagy, we evaluated the subcellular localization of the mammalian target of rapamycin (mTOR). Under normal conditions the active mTOR complex 1 (mTORC1) is located at the lysosomal membrane and colocalizes with lysosomal membrane protein LAMP1. Imaging analysis showcased a 5-fold loss of co-localization of the LAMP1-mTOR complex in both *CTNS*^−/−^ and *CTNS^Patient^* kidney tubules (Figure 7), when compared to the healthy kidney tubules. These results indeed suggest the inactivation of mTOR, which will in turn activate autophagy.

### 3.5. Cystinotic Kidney Tubules Present Metabolic Impairment

To gain in-depth knowledge on the intracellular molecular pathways affected in the cystinotic kidney tubules, we evaluated the metabolomic profile of 100 key metabolites in all three models. Principal component analysis (PCA) of the metabolites measured show that *CTNS*^−/−^ and *CTNS^Patient^* kidney tubules account for most of the variability of the dataset (74.2%); Figure 8A). To explore which metabolites and pathways are directly linked to a cystinotic phenotype, the top 60 most differentially expressed metabolites in cystinotic *vs* healthy kidney tubules were plotted together in a heatmap (Figure 8B). Significantly altered metabolites are either up or downregulated, such as the accumulation of alanine and glucose, and the loss of threonine and glutamine in the cystinotic models when compared to the healthy kidney tubules. Interestingly, despite the uniqueness of the genetic background of the *CTNS*^−/−^ and the *CTNS^Patient^* models, the metabolic profiling appears to be similar when compared to the healthy kidney tubules, indicating that these metabolites are directly linked to *CTNS* loss rather than to differences in the genetic background of the cells.

To elucidate the molecular pathways altered in cystinosis due to complete *CTNS* loss, we performed a pathway analysis that revealed the pathways significantly correlated (*p* < 0.05) with *CTNS* loss: ubiquinone biosynthesis, phenylalanine, tyrosine and tryptophan biosynthesis, arginine biosynthesis, cystine and methionine metabolism, D-Glutamine and D-glutamate metabolism, alanine, aspartate and glutamate metabolism, and the TCA cycle (tricarboxylic acid cycle), in decreasing order of impact. In line with previous studies, we found α-ketoglutarate (α-KG) to be among the key metabolites altered when comparing cystinotic to healthy kidney tubules (Figure 9B). Alpha-Ketoglutarate dehydrogenase is the enzyme encoded by the *AKGDH* gene and is responsible for the degradation of α-KG in the cytosol. We found a significant reduction in *AKGDH* expression in both cystinotic models (Figure 9A), which correlates directly with the cytosolic accumulation of a-KG (Figure 9B).

**Figure 6 cells-11-00177-f006:**
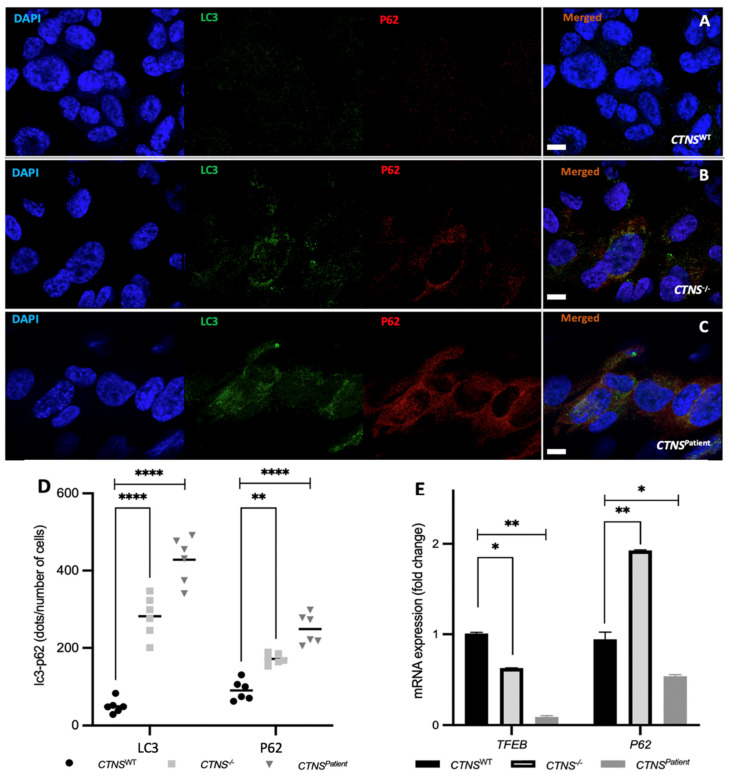
Autophagic markers in kidney tubules. Images obtained with a confocal microscope at 63X of the ciPTEC *CTNS^WT^* (**A**), *CTNS^−/−^* (**B**), and *CTNS^Patient^* (**C**) kidney tubules. Image quantification analysis showed a significant increase of the autophagy markers LC3 and p62 in the cystinotic 3D models when compared to the healthy control (**D**). Real-time PCR quantification also showed a significant impairment of the autophagy-related genes TFEB and p62 in the cystinotic models when compared to the healthy control (**E**). In blue: nuclei, in green: LC3 protein, in red: p62 protein. Scale bar: 10 μm. One-way ANOVA statistical analysis was performed (N = 3; * *p*-value < 0.05; ** *p*-value < 0.01; **** *p*-value < 0.0001).

**Figure 7 cells-11-00177-f007:**
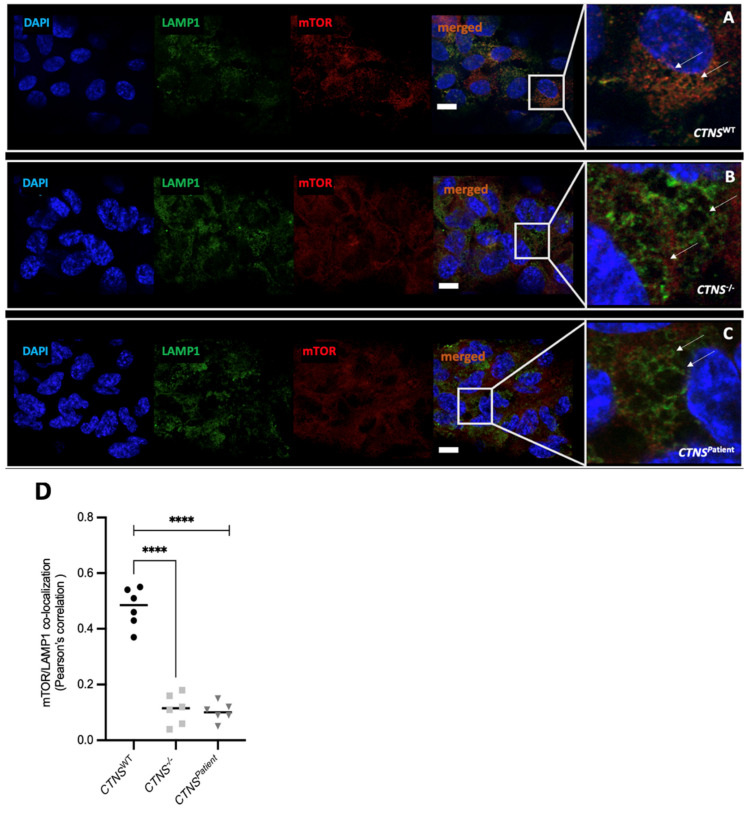
mTOR and LAMP1 staining and colocalization in kidney tubules. Confocal microscopy images taken at 63X of the ciPTEC *CTNS**^WT^* (**A**), *CTNS*^−/−^ (**B**), and *CTNS**^Patient^* (**C**) kidney tubules. Image analysis quantification showed co-localization of the mTOR/LAMP1 complex in the healthy control and loss of co-localization in both cystinotic models (**D**), suggesting an impaired autophagosome-lysosome trafficking. In blue: DAPI (nuclei), in green: LAMP1 protein, in red: mTOR protein. Scale bar: 10 μm. One-way ANOVA statistical analysis was performed (N = 3; **** *p*-value < 0.0001).

**Figure 8 cells-11-00177-f008:**
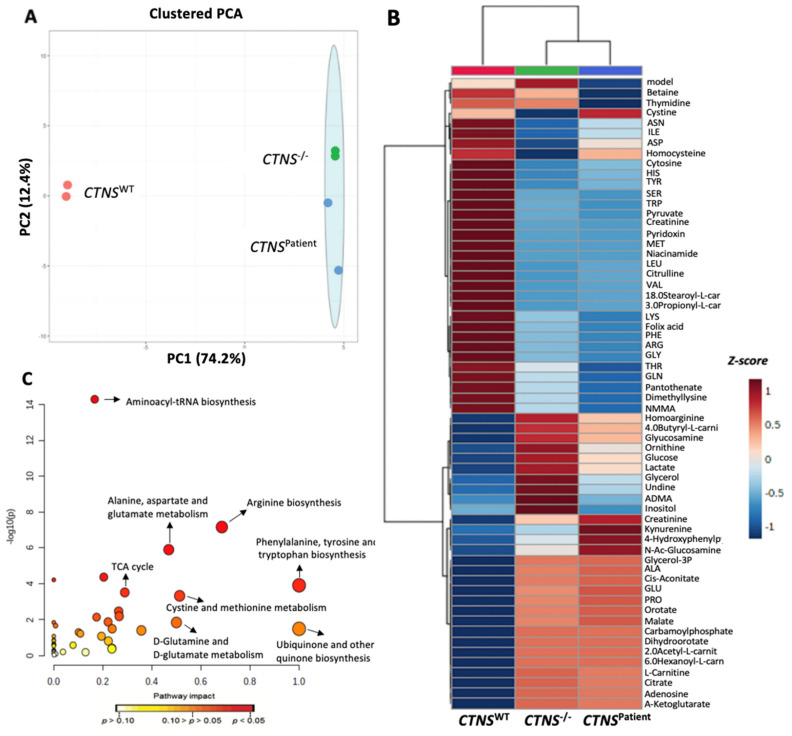
Metabolomic profiling of kidney tubules. Principal component analysis (PCA) of the ciPTEC *CTNS*^WT^, *CTNS*^−/−^, and *CTNS**^Patient^* kidney tubule models based on the metabolites set measured. In the plot, the individual dots represent one biological repeat, and dots of the same color are the same experimental group (healthy (*CTNS^WT^*, *CTNS*^−/−^ and *CTNS**^Patient^* kidney tubules) (**A**). Heatmap analysis of the top 60 metabolites differentially expressed in healthy and cystinotic kidney tubule models. Every row represents a different metabolite and its associated Z-score. Significantly increased metabolites (*p* < 0.01) are displayed in red, and significantly decreased metabolites (*p* < 0.01) are displayed in blue (**B**). Global pathway enrichment analysis of the metabolic pathways differentially expressed in the *CTNS*^−/−^ compared to the healthy kidney tubules. The larger the circles and the further they appear from the y-axis, the higher the impact of that pathway in the *CTNS*^−/−^ kidney tubules (**C**). Data was normalized to the median intensity. Data analysis was performed using univariate and multivariate analysis (N = 2; *p*-values < 0.05 were considered significant).

**Figure 9 cells-11-00177-f009:**
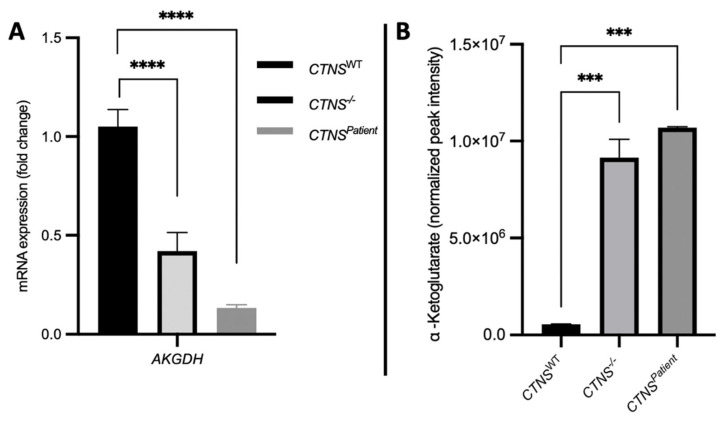
Cystinotic kidney tubules accumulate α-KG. Real-time PCR quantification show a significant 45% and 89% reduction (*CTNS*^−/−^ and *CTNS**^Patient^* kidney tubules, respectively) of the *AKGDH* gene (**A**). Cystinotic kidney tubules significantly accumulate α-KG in the cytoplasm (**B**). One-way ANOVA statistical analysis was performed (N = 3; *** *p*-value < 0.001; **** *p*-value < 0.0001).

## 4. Discussion

In this work, we developed and characterized 3-dimensional bioengineered proximal tubule models for studying nephropathic cystinosis. We compared two cystinotic kidney tubule models to a healthy proximal tubule. To avoid misleading results of the cystinotic phenotype due to the different genetic background between the cystinotic patient donor and the healthy donor, we included an isogenic cell line (*CTNS*^−/−^), which was previously developed in-house using CRISPR/Cas9 from the healthy control cell line [9].

In comparison to the conventional 2-dimensional cultures, our HFM approach allows the cells to grow and develop in structures that mimic a tubule. All three cell lines cultured were able to proliferate and maturate on the HFM and exhibit polarization markers, including the cilia and the Na^+^/K^+^-ATPase at the apical and basolateral side, respectively. This 3-dimensional tubular structure is crucial for studies that require molecular transport, since both the apical and the basolateral sides of the cells are accessible. Previous studies already demonstrated how the geometry of the culture surface affects the gene expression patterns and molecular pathways of the cells [38], including the kidney tubular cells [39]. Most studies reported a more mature cell differentiation when cultured in a 3-dimensional, microfluidics platform as compared to 2-dimensional cultures [9,40]. 

Cystinosis is mainly characterized by the loss or impairment of the *CTNS* gene, which leads to the intralysosomal accumulation of cystine [1]. Our results show a significant reduction (60%) and a total loss of *CTNS* gene expression in *CTNS*^−/−^ and *CTNS^Patient^* kidney tubules, respectively. Consequently, the cystine accumulation found in both cystinotic models are more profound when compared to 2-dimensional cell cultures, especially in the *CTNS^Patient^* kidney tubules, which was 3-fold higher [9], and mimics the diagnostic testing results in the white blood cells of cystinotic patients [41]. We can hypothesize that growing these cells in the HFM 3-dimensional configuration enhances the cystinotic phenotype caused by the complete loss of cystinosin in these cells. 

Both cystinotic models exhibit a loss of organization of the cell monolayer and a significantly higher leakage when compared to the healthy kidney tubules. Clinically, the renal presentation of cystinosis includes a dilated and atrophic proximal tubule [42], a direct consequence of a loss of monolayer tightness and epithelial organization [43]. This phenotype was clearly observed in our cystinotic kidney tubules and was more profound in the *CTNS^Patient^* model, which harbors the 57kb deletion in *CTNS*. Furthermore, *in vivo* cystinosis studies reported abnormal levels of autophagy [44,45], with changes in the expression of key autophagy genes such as *TFEB*, *LC3II,* and *SQSTM1*. Our results show a significant increase of p62 and LC3 protein in both cystinotic tubules, consistent with increased autophagy in both models of cystinosis. Furthermore, our data reveals that the mTOR/LAMP1protein complex is dissociated in both cystinotic kidney tubule models when compared to the heathy control. The inactivation of mTOR is expected to activate the translocation of the transcription factor TFEB to the nucleus, which in turn will downregulate its own mRNA levels, which is also what we observed in our system. The mRNA levels of p62 were more puzzling. In line with increased protein levels, the expression of p62 was increased in the *CTNS*^−/−^ tubules when compared to the *CTNS^WT^*. In the *CTNS^Patient^* tubules, on the other hand, p62 mRNA levels were consistently down. This may be due to a (slight) difference in autophagic flux. We previously found that, after blocking, the autophagic flux with bafilomycin the autophagy markers LC3 and p63 would rise, and to a higher extend in *CTNS*^−/−^ cells compared to *CTNS^Patient^* cells [9]. In this case the *CTNS*^−/−^ tubules also have to produce and degrade higher levels of autophagic cargo (including p62), whereas in the *CTNS^Patient^* cells the cargo degradation may be delayed. As p62 is accumulating and not efficiently degraded there may be no need for the *CTNS^Patient^* cells to keep producing more p63 mRNA (resulting in lower mRNA levels). In the literature there is also conflicting evidence to what extent the increase in autophagy seen in cystinosis is accompanied by a block in autophagy or that the autophagic flux is actually increased. This also suggests that the outcome of this assay may depend on the model system used and other factors, such as genetic background, may play a role. Overall, our data on vesicle trafficking impairment is consistent with the phenotype previously observed in cystinosis [4,43,46,47].

Despite the gap in understanding the link between cystine accumulation and its implication in the molecular pathways leading to nephropathy, cystinosis has been extensively described as a metabolic disease [48,49]. Our metabolomic analysis displays the impact of *CTNS* loss in the intra-cellular pathways in cystinotic kidney tubules, showcasing that cystinosis is not only a lysosomal storage disease, but a condition that significantly affects many metabolic processes such as glucose, glutamine, and alanine metabolism and the TCA cycle. Some of these pathways are directly related while others are not, but the accumulation of cystine seems to be the key element in such a metabolic impairment. As previously mentioned, renal Fanconi syndrome appears in early stages in cystinotic patients. The clinical presentation includes the loss of amino acids and proteins, which leads to proteinuria/aminoaciduria, glycosuria, and hypophosphatemia/hyperphosphaturia [50,51]. Therefore, kidney tubules are a powerful instrument to study in depth the metabolic pathways underlying cystinosis and, hence, offer a promising platform to further study the metabolic and proteomic disturbances in renal Fanconi syndrome.

Autophagy is a digestion process that fuels the cells with essential aminoacids that will eventually enter the TCA cycle. One metabolite involved in the TCA cycle and that has been described previously in increased autophagy is α-KG, which appears to play a role in key pathways in cystinosis, such as glucose, glutamine, and alanine metabolism. Numerous studies have described the close relationship between α-KG and cystine [52,53], including its downstream effect on oxidative stress and role in the glucose, glutamine and other aminoacids metabolism. Moreover, under healthy conditions, α-KG is degraded by the mitochondrial enzyme *AKGDH*, which we observed significantly reduced in both cystinotic kidney tubule models. Mitochondrial dysfunction has also been reported for cystinotic cell lines [54], which could lead to the downregulation of *AKGDH* and, subsequently, the induction of α-KG accumulation. Both cystinotic kidney tubules demonstrated a significant accumulation of α-KG and a downregulation of *AKGDH* when compared to the healthy kidney tubules. The metabolic analysis of the cystinotic kidney tubules showcased the similarities between the isogenic *CTNS*^−/−^ and *CTNS^Patient^* 3D models. Similarly, the differences in the metabolic pathways between the cystinosis models and the healthy bioengineered kidney tubules were also better represented in 3D than what has been previously seen in 2-dimensional cell culture studies [9]. 

The enhancement of the hallmarks of the disease or the better representation of the clinical condition in 3D models is a recurrent observation in the literature for many disorders. For instance, reproducing the cell-to-extracellular matrix interaction, recapitulating the expression of apoptosis-related genes, the increase of drug sensitivity, and mimicking tumor invasion processes are some of the traits that have shown to be better represented in 3D cancer models [55,56,57]. 3D liver disease models have also been reported advantageous over 2D cultures since they allow for the study of biliary excretion of metabolites, the functional establishment of cell polarity, and crucial processes in liver disease such as inflammation and fibrosis [58,59,60]. Additionally, having 3D cultures made it possible to study the metabolic interaction between diseased lung cells and cancer-associated fibroblasts [61], but many more examples have been published. 

In this study, we aimed to establish the added benefit of culturing cystinotic cells in 3-dimensions on an HFM. In addition to the phenotypical characteristics that these cells show in a 2-dimensional setting, we could now also observe differences in cell organization and the alignment of cells along the direction of the fiber. Furthermore, growing the cells on a porous membrane allowed us to perform a leakage assay, which showed that the monolayer of the cystinotic cells was less tight compared to control cells. That said, culturing the cells on an HFM also comes with some drawbacks. You need more cells because only part of the cells will adhere the membrane in contrast with 2-dimensional culturing where all the seeded cells will end up in the well. Additionally, care must be taken to not disrupt the kidney tubule monolayer during handling, and imaging must be conducted with one fiber at the time, making this approach more labor intensive and less suitable for high throughput applications. Furthermore, assays that require a larger number of cells (such as western blotting) remain challenging. Therefore, growing cells as a kidney tubule may not be the best option for all types of analysis, but rather adds to the toolbox of methods available for the study of proximal tubule diseases, especially when cell transport and cellular organization are involved. 

In summary, our bioengineered kidney tubule models mimic the pathophysiology and the morphological abnormalities of nephropathic cystinosis better than the 2-dimensional cell culture models available to study this disease. In addition, the HFM platform offers a complex system that allows for perfusion, which facilitates the assessment of metabolites transport and encourages drug testing in a proximal tubule-imitating architecture. Hence, transitioning from 2-dimensional cell cultures to complex, microphysiological models that better mimic the disease is crucial to advance our understanding of the molecular mechanisms of the disease of interest and paves the way to personalized medicine. 

## Figures and Tables

**Figure 1 cells-11-00177-f001:**
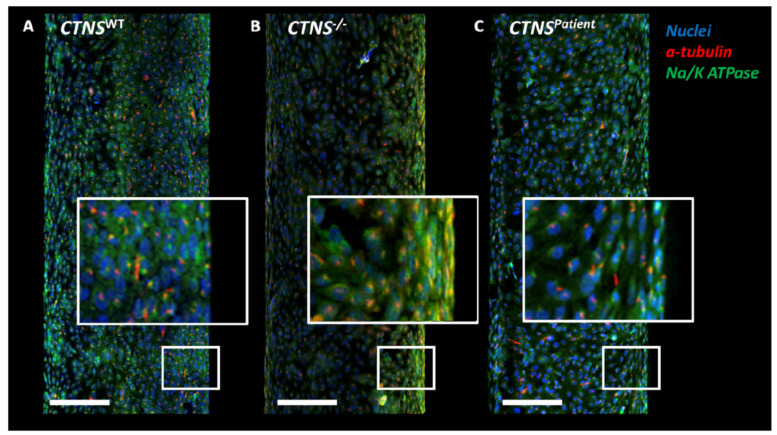
Confocal images of healthy and cystinotic ciPTECs kidney tubules. After maturation of 7 days, the control ciPTEC (**A**) form an organized 3-dimensional structure including primary cilia. Both cystinotic kidney tubule models (**B**,**C**) show visible holes in the monolayer when compared to the healthy proximal tubule model (**A**). The close-up images represent a 3-fold zoom increase. In blue: DAPI (nuclei staining), in red: α-tubulin staining, in green: Na^+^/K^+^-ATPase. Scale bar: 100 μm.

**Figure 2 cells-11-00177-f002:**
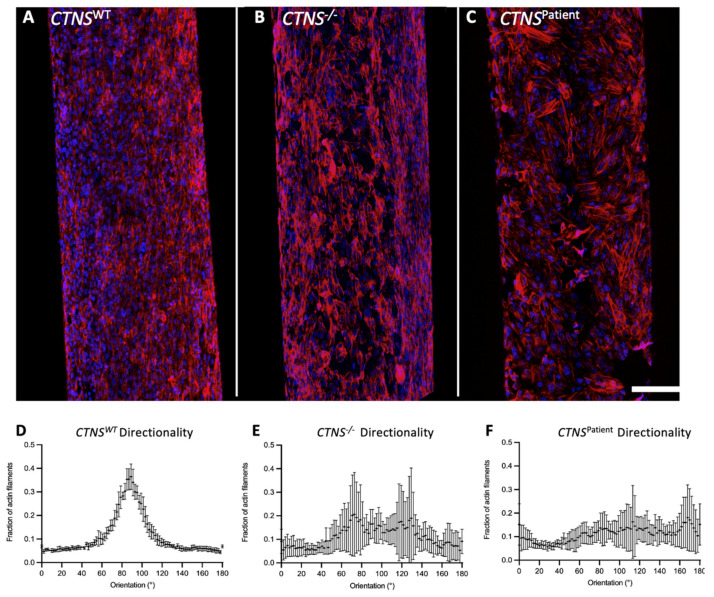
Cellular organization of kidney tubules. Kidney tubules were stained using phalloidin to show the organization of actin filaments in ciPTEC *CTNS^WT^* (**A**), *CTNS*^−/−^ (**B**), and *CTNS^Patient^* (**C**). Both cystinotic models (**B**,**C**) present a disrupted cell monolayer with visible holes along the membrane when compared to the control cells (**A**), which is also seen in the image directionality quantification (**D**–**F**). In blue: DAPI (nuclei staining), in red: Phalloidin (binds to actin filaments). Directionality analysis was performed in three biological replicates. Scale bar: 100 μm.

**Figure 3 cells-11-00177-f003:**
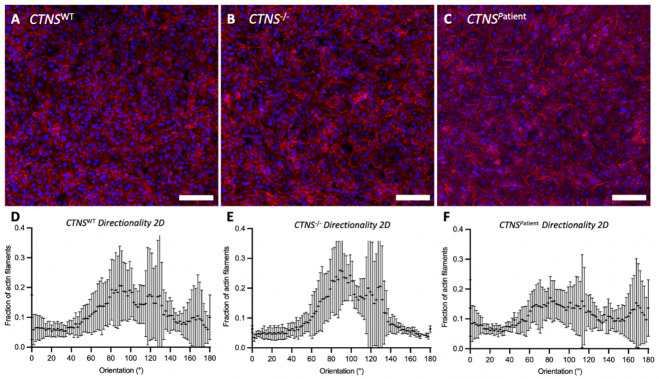
Cellular organization in 2D cell culture. Cells were stained using phalloidin to show the organization of actin filaments in ciPTEC *CTNS*^WT^ (**A**), *CTNS*^−/−^ (**B**), and *CTNS*^Patient^ (**C**). Image directionality analysis revealed loss of organization in all the cell lines when cultured in 2D (**D**–**F**). In blue: DAPI (nuclei staining), in red: Phalloidin (binds to actin filaments). Directionality analysis was performed in two biological replicates Scale bar: 100 μm.

**Figure 4 cells-11-00177-f004:**
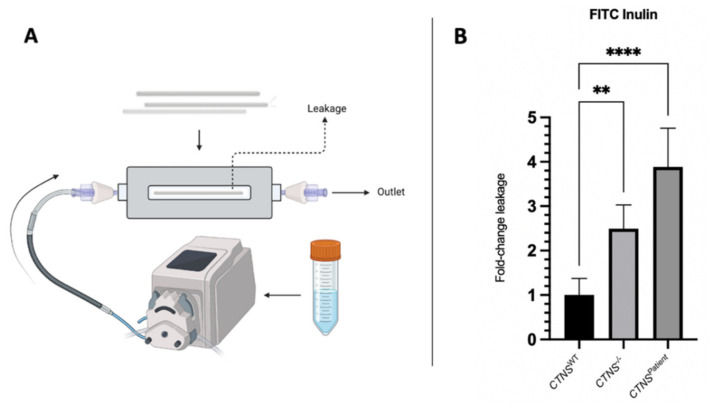
FITC inulin leakage assay in kidney. (**A**) Graphical presentation of the experimental set-up for the assessment of the monolayer leakage. The kidney tubules were secured into a 3-D printed chamber and connected to a pump with an in-let and out-let needle. Fluorescent FITC-inulin solution was perfused through the inside of the fiber and after 10 min, the solution that leaked through the kidney tubules to the extraluminal compartment was collected and measured. (**B**) Leakage of the healthy and cystinotic kidney tubules is expressed in fold change compared to the healthy control and normalized to a double-coated but unseeded HFM. One-way ANOVA statistical analysis was performed (N = 3; ** *p*-value < 0.01; **** *p*-value < 0.0001).

**Figure 5 cells-11-00177-f005:**
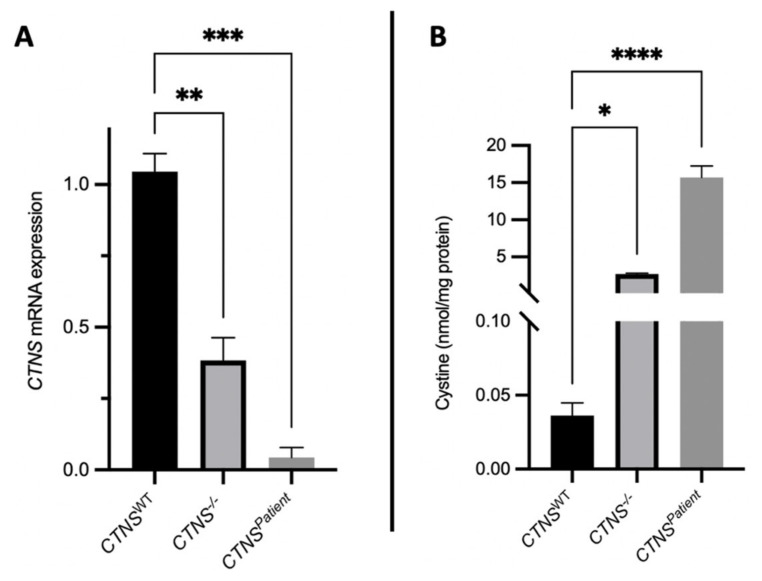
Cystine accumulation and *CTNS* expression in kidney tubules. Real-time PCR quantification showed significant reduction of *CTNS* gene expression in both cystinotic tubule models when compared to the healthy control (**A**), which led to an increase in cystine accumulation in the cystinotic kidney tubules (**B**). One-way ANOVA statistical analysis was performed (N = 3; * *p*-value < 0.05; ** *p*-value < 0.01; *** *p*-value < 0.001; **** *p*-value < 0.0001).

**Table 1 cells-11-00177-t001:** Primer sets used for quantification by Real-time PCR.

Gene	Forward Primer (5′-3′)	Reverse Primer (5′-3′)
*CTNS*	AGCTCCCCGATGAAGTTGTG	GTCAGGTTCAGAGCCACGAA
*TFEB*	GCAGTCCTACCTGGAGAATC	GTGGGCAGCAAACTTGTTCC
*SQSTM1* (*p62*)	CTGAGCTCTGCCTCTTCCAG	GACAGGAGGAACAGTGAGGC
*AKGDH*	GATCTGGACTCCTCCGTGCC	ATCTCCCGCAGAGGAAGTGC
*RPS-13*	GCTCTCCTTTCGTTGCCTGA	ACTTCAACCAAGTGGGGACG

## Data Availability

The raw data for the metabolomics analysis is currently being uploaded to a public data depository.

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
