# Peer review of "Bioengineered Cystinotic Kidney Tubules Recapitulate a Nephropathic Phenotype"

_cells, 2022, doi:10.3390/cells11010177_

Round 1
Reviewer 1 Report
The manuscript by Garvi et al., describes the characterization of kidney tubules as a new 3D model to study nephropathic cystinosis. The authors culture ciPTECs on hollow fiber membranes to establish mature kidney tubules that form an epithelial monolayer with the cystinotic tubules exhibiting a degree of disruption to this epithelial layer. They show that these tubules have a number of well characterized cystinotic phenotypes such as cystine accumulation, altered autophagy and vesicle trafficking, metabolic pathway disturbances, in particular increased levels of alpha-ketoglutarate.
The authors have presented a new model which will be a welcome addition to the field and are encouraged to consider the following comments to improve the report:
- The Figure 1B image needs to be replaced. The close-up image appears blurry, making it difficult to decipher the markers which are clear in panels A and C.
- Please add to materials and methods how the quantification of puncta in Figs 6 and 7 was performed.
- In cystinotic cells frequently a perinuclear accumulation of lysosomes is present, is this the case in the tubules?
- Do these tubules express the endocytic receptor megalin an cubilin?
- Is the endocytic uptake normal or abnormal in these diseased cells? Can you perform an experiment with FITC-BSA to assess?
- As well as mTOR localization what are the levels of mTOR protein in the CTNS-/- tubules?
- Is TFEB translocated normally in these tubules?
- Is autophagic flux disrupted in these tubules, can you perform a LC3I/LC3II ratio experiment?
- It is interesting that p62 gene expression is increased in the KO line compared to the patient line, and a similar phenotype was observed previously in cell lines, do the authors have an idea of why this would be? An explanation should be added to the discussion.
- Are apoptosis and ROS increased/decreased in the tubules
- Does treatment with Bicalutamide and cysteamine correct the phenotypes observed in tubules?
Minor comments:
Line 269 – figure 3A should be figure 4A
Line 366 – ‘Images under the confocal microscope’ please re-word
Author Response
Thank you for your kind suggestions and critical evaluation of our manuscript. The point by point reply to your questions can be found in the attached file.

Reviewer 2 Report
Garvi et al present "Bioengineered cystinosis kidney tubules recapitulate a nephropathic phenotype" which outlines the study of an immortalized cell line with and without CTNS as well as a patient cell line with cystinosis. Specifically, they look at the role of the HFM and how this can help to better recapitulate biological observations as compared to a 2D system.
Overall, the manuscript is well written, clear and I have primarily minor comments.
1) would recommend just clarifying that (eg 2.10 would say that these are "biological triplicates, including at least 3 technical replicates"
2) section 3.4, can the authors comment on the differential in P62 - this was missing in the discussion. could this be due to how the cells were immortalized? is there any concern that P62 is close to where CTNS is and as such may be affected with Cas9 cutting of CTNS?
3) Figure 8b - the labels are hard to read and would recommend fixing that
4) In the discussion
a) can the authors comment if there was a difference in the ability to continue to culture the cells on HFM as compared to 2d?
b) throughout the discussion, would recommend that the authors work to clarify when protein by proxy (eg IF) is used vs qRT-PCR. For example, immunoblots (eg Westerns) were not performed to confirm protein downregulation. And then in lines 438/9 the term "loss of CTNS expression" leaves a bit of lack of clarity as this was based on RNA rather than protein. Would recommend going through the discussion to clarify these to be consistent with the data presented.
Author Response

(The authors gave the same response as above.)

Round 2
Reviewer 1 Report
The authors have responded adequately to all requests for changes and explained where applicable why a request was not possible, overall improving the quality of the manuscript.
Suggestion:
I wonder should the authors revise the use of the term 'proximal' tubule as without showing detection of proximal tubule-specific markers in these 3D tubules I don't think the authors can reliably state that the tubules are proximal despite the fact that they are derived from an immortalised PTEC line.
I would suggest using the term 'tubules' until full characterisation is complete.
Author Response
We understand the concern of the reviewer with regards to the term “proximal tubule” in this study. Nevertheless, due to our experience with culturing ciPTECs on hollow fiber membranes and the previous studies we performed using these 3-D systems, we are confident that we gathered enough evidence to refer to these systems as “proximal kidney tubules”.
CiPTEC14.4 is a human conditionally immortalized proximal tubular epithelial cell line. Since its development and characterization by Wilmer et al. in 2010 [1], this cell line has been used in multiple studies within our group but have also been widely used by many other groups. This resulted in more than 60 publications in PubMed. Since its immortalization, the cell line has shown to stably express the proximal tubule phenotype over a very high number of passages (>60). Our group has extensive experience conducting research with ciPTEC lines, and great efforts have been put in the last years into creating and optimizing these proximal kidney tubule systems. Among the functional studies, we showed that these kidney tubules are capable of organic cation transporter (OCT)-mediated ASP+ transport, albumin reabsorption, organic anion secretion, and secretion of immune modulators upon an inflammatory response [2-5].
Some of this information was indeed lacking from the manuscript and we have updated the introduction (line 79-83).
We would like to thank this reviewer again for taking the time to provide feedback and improve our manuscript as a whole.
Kind regards also on behalf of all co-authors,
Manoe Janssen
References
[1] Wilmer, M. J., Saleem, M. A., Masereeuw, R., Ni, L., van der Velden, T. J., Russel, F. G., ... & Levtchenko, E. N. (2010). Novel conditionally immortalized human proximal tubule cell line expressing functional influx and efflux transporters. Cell and tissue research, 339(2), 449-457
[2] Jansen, J., De Napoli, I. E., Fedecostante, M., Schophuizen, C. M. S., Chevtchik, N. V., Wilmer, M. J., ... & Masereeuw, R. (2015). Human proximal tubule epithelial cells cultured on hollow fibers: living membranes that actively transport organic cations. Scientific reports, 5(1), 1-12
[3] Schophuizen, C. M., De Napoli, I. E., Jansen, J., Teixeira, S., Wilmer, M. J., Hoenderop, J. G., ... & Stamatialis, D. (2015). Development of a living membrane comprising a functional human renal proximal tubule cell monolayer on polyethersulfone polymeric membrane. Acta biomaterialia, 14, 22-32
[4] Chevtchik, N. V., Mihajlovic, M., Fedecostante, M., Bolhuis‐Versteeg, L., Sastre Toraño, J., Masereeuw, R., & Stamatialis, D. (2018). A bioartificial kidney device with polarized secretion of immune modulators. Journal of tissue engineering and regenerative medicine, 12(7), 1670-1678
[5] Englezakis, A., Gozalpour, E., Kamran, M., Fenner, K., Mele, E., & Coopman, K. (2021). Development of a hollow fibre-based renal module for active transport studies. Journal of Artificial Organs, 1-12